# On the Computational Efficiency of Training Neural Networks

**Roi Livni**
The Hebrew University
roi.livni@mail.huji.ac.il

**Shai Shalev-Shwartz**
The Hebrew University
shais@cs.huji.ac.il

**Ohad Shamir**
Weizmann Institute of Science
ohad.shamir@weizmann.ac.il

## Abstract

It is well-known that neural networks are computationally hard to train. On the other hand, in practice, modern day neural networks are trained efficiently using SGD and a variety of tricks that include different activation functions (e.g. ReLU), over-specification (i.e., train networks which are larger than needed), and regularization. In this paper we revisit the computational complexity of training neural networks from a modern perspective. We provide both positive and negative results, some of them yield new provably efficient and practical algorithms for training certain types of neural networks.

## 1 Introduction

One of the most significant recent developments in machine learning has been the resurgence of "deep learning", usually in the form of artificial neural networks. A combination of algorithmic advancements, as well as increasing computational power and data size, has led to a breakthrough in the effectiveness of neural networks, and they have been used to obtain very impressive practical performance on a variety of domains (a few recent examples include [17, 16, 24, 10, 7]).

A neural network can be described by a (directed acyclic) graph, where each vertex in the graph corresponds to a neuron and each edge is associated with a weight. Each neuron calculates a weighted sum of the outputs of neurons which are connected to it (and possibly adds a bias term). It then passes the resulting number through an activation function $\sigma : \mathbb{R} \to \mathbb{R}$ and outputs the resulting number. We focus on feed-forward neural networks, where the neurons are arranged in layers, in which the output of each layer forms the input of the next layer. Intuitively, the input goes through several transformations, with higher-level concepts derived from lower-level ones. The depth of the network is the number of layers and the size of the network is the total number of neurons.

From the perspective of statistical learning theory, by specifying a neural network architecture (i.e. the underlying graph and the activation function) we obtain a hypothesis class, namely, the set of all prediction rules obtained by using the same network architecture while changing the weights of the network. Learning the class involves finding a specific set of weights, based on training examples, which yields a predictor that has good performance on future examples. When studying a hypothesis class we are usually concerned with three questions:

1. *Sample complexity:* how many examples are required to learn the class.
2. *Expressiveness:* what type of functions can be expressed by predictors in the class.
3. *Training time*: how much computation time is required to learn the class.

For simplicity, let us first consider neural networks with a threshold activation function (i.e. $\sigma(z) = 1$ if $z > 0$ and 0 otherwise), over the boolean input space, $\{0, 1\}^d$, and with a single output in $\{0, 1\}$. The sample complexity of such neural networks is well understood [3]. It is known that the VC dimension grows linearly with the number of edges (up to log factors). It is also easy to see that no matter what the activation function is, as long as we represent each weight of the network using

a constant number of bits, the VC dimension is bounded by a constant times the number of edges. This implies that empirical risk minimization - or finding weights with small average loss over the training data - can be an effective learning strategy from a statistical point of view.

As to the expressiveness of such networks, it is easy to see that neural networks of depth 2 and sufficient size can express all functions from $\{0,1\}^d$ to $\{0,1\}$. However, it is also possible to show that for this to happen, the size of the network must be exponential in $d$ (e.g. [19, Chapter 20]). Which functions can we express using a network of polynomial size? The theorem below shows that all boolean functions that can be calculated in time $O(T(d))$, can also be expressed by a network of depth $O(T(d))$ and size $O(T(d)^2)$.

**Theorem 1.** *Let $T : \mathbb{N} \to \mathbb{N}$ and for every d, let $\mathcal{F}_d$ be the set of functions that can be implemented by a Turing machine using at most $T(d)$ operations. Then there exist constants $b, c \in \mathbb{R}_+$ such that for every d, there is a network architecture of depth $c\,T(d) + b$, size of $(c\,T(d) + b)^2$, and threshold activation function, such that the resulting hypotesis class contains $\mathcal{F}_d$.*

The proof of the theorem follows directly from the relation between the time complexity of programs and their circuit complexity (see, e.g., [22]), and the fact that we can simulate the standard boolean gates using a fixed number of neurons.

We see that from the statistical perspective, neural networks form an excellent hypothesis class; On one hand, for every runtime $T(d)$, by using depth of $O(T(d))$ we contain all predictors that can be run in time at most $T(d)$. On the other hand, the sample complexity of the resulting class depends polynomially on $T(d)$.

The main caveat of neural networks is the training time. Existing theoretical results are mostly negative, showing that successfully learning with these networks is computationally hard in the worst case. For example, neural networks of depth 2 contain the class of intersection of halfspaces (where the number of halfspaces is the number of neurons in the hidden layer). By reduction to $k$-coloring, it has been shown that finding the weights that best fit the training set is NP-hard ([9]). [6] has shown that even finding weights that result in close-to-minimal empirical error is computationally infeasible. These hardness results focus on *proper* learning, where the goal is to find a nearly-optimal predictor with a fixed network architecture $A$. However, if our goal is to find a good predictor, there is no reason to limit ourselves to predictors with one particular architecture. Instead, we can try, for example, to find a network with a different architecture $A'$, which is almost as good as the best network with architecture $A$. This is an example of the powerful concept of *improper* learning, which has often proved useful in circumventing computational hardness results. Unfortunately, there are hardness results showing that even with improper learning, and even if the data is generated exactly from a small, depth-2 neural network, there are no efficient algorithms which can find a predictor that performs well on test data. In particular, [15] and [12] have shown this in the case of learning intersections of halfspaces, using cryptographic and average case complexity assumptions. On a related note, [4] recently showed positive results on learning from data generated by a neural network of a certain architecture and randomly connected weights. However, the assumptions used are strong and unlikely to hold in practice.

Despite this theoretical pessimism, in practice, modern-day neural networks are trained successfully in many learning problems. There are several tricks that enable successful training:

- *Changing the activation function:* The threshold activation function, $\sigma(a) = \mathbf{1}_{a>0}$, has zero derivative almost everywhere. Therefore, we cannot apply gradient-based methods with this activation function. To circumvent this problem, we can consider other activation functions. Most widely known is a sigmoidal activation, e.g. $\sigma(a) = \frac{1}{1+e^a}$, which forms a smooth approximation of the threshold function. Another recent popular activation function is the rectified linear unit (ReLU) function, $\sigma(a) = \max\{0, a\}$. Note that subtracting a shifted ReLU from a ReLU yields an approximation of the threshold function, so by doubling the number of neurons we can approximate a network with threshold activation by a network with ReLU activation.

- *Over-specification:* It was empirically observed that it is easier to train networks which are larger than needed. Indeed, we empirically demonstrate this phenomenon in Sec. 5.

- *Regularization:* It was empirically observed that regularizing the weights of the network speeds up the convergence (e.g. [16]).

The goal of this paper is to revisit and re-raise the question of neural network's computational efficiency, from a modern perspective. This is a challenging topic, and we do not pretend to give any definite answers. However, we provide several results, both positive and negative. Most of them are new, although a few appeared in the literature in other contexts. Our contributions are as follows:

- We make a simple observation that for sufficiently over-specified networks, global optima are ubiquitous and in general computationally easy to find. Although this holds only for extremely large networks which will overfit, it can be seen as an indication that the computational hardness of learning does decrease with the amount of over-specification. This is also demonstrated empirically in Sec. 5.

- Motivated by the idea of changing the activation function, we consider the quadratic activation function, $\sigma(a) = a^2$. Networks with the quadratic activation compute polynomial functions of the input in $\mathbb{R}^d$, hence we call them polynomial networks. Our main findings for such networks are as follows:

  - Networks with quadratic activation are as expressive as networks with threshold activation.
  - Constant depth networks with quadratic activation can be learned in polynomial time.
  - Sigmoidal networks of depth 2, and with $\ell_1$ regularization, can be approximated by polynomial networks of depth $O(\log \log(1/\epsilon))$. It follows that sigmoidal networks with $\ell_1$ regularization can be learned in polynomial time as well.
  - The aforementioned positive results are interesting theoretically, but lead to impractical algorithms. We provide a practical, provably correct, algorithm for training depth-2 polynomial networks. While such networks can also be learned using a linearization trick, our algorithm is more efficient and returns networks whose size does not depend on the data dimension. Our algorithm follows a forward greedy selection procedure, where each step of the greedy selection procedure builds a new neuron by solving an eigenvalue problem.
  - We generalize the above algorithm to depth-3, in which each forward greedy step involves an efficient approximate solution to a tensor approximation problem. The algorithm can learn a rich sub-class of depth-3 polynomial networks.
  - We describe some experimental evidence, showing that our practical algorithm is competitive with state-of-the-art neural network training methods for depth-2 networks.

## 2  Sufficiently Over-Specified Networks Are Easy to Train

We begin by considering the idea of over-specification, and make an observation that for sufficiently over-specified networks, the optimization problem associated with training them is generally quite easy to solve, and that global optima are in a sense ubiquitous. As an interesting contrast, note that for very small networks (such as a single neuron with a non-convex activation function), the associated optimization problem is generally hard, and can exhibit exponentially many local (non-global) minima [5]. We emphasize that our observation only holds for extremely large networks, which will overfit in any reasonable scenario, but it does point to a possible spectrum where computational cost decreases with the amount of over-specification.

To present the result, let $X \in \mathbb{R}^{d,m}$ be a matrix of $m$ training examples in $\mathbb{R}^d$. We can think of the network as composed of two mappings. The first maps $X$ into a matrix $Z \in \mathbb{R}^{n,m}$, where $n$ is the number of neurons whose outputs are connected to the output layer. The second mapping is a linear mapping $Z \mapsto WZ$, where $W \in \mathbb{R}^{o,n}$, that maps $Z$ to the $o$ neurons in the output layer. Finally, there is a loss function $\ell : \mathbb{R}^{o,m} \to \mathbb{R}$, which we'll assume to be convex, that assesses the quality of the prediction on the entire data (and will of course depend on the $m$ labels). Let $V$ denote all the weights that affect the mapping from $X$ to $Z$, and denote by $f(V)$ the function that maps $V$ to $Z$. The optimization problem associated with learning the network is therefore $\min_{W,V} \ell(W f(V))$.

The function $\ell(W f(V))$ is generally non-convex, and may have local minima. However, if $n \geq m$, then it is reasonable to assume that $\text{Rank}(f(V)) = m$ with large probability (under some random choice of $V$), due to the non-linear nature of the function computed by neural networks[1]. In that case, we can simply fix $V$ and solve $\min_W \ell(W f(V))$, which is computationally tractable as $\ell$ is

assumed to be convex. Since $f(V)$ has full rank, the solution of this problem corresponds to a global optima of $\ell$, and hence to a global optima of the original optimization problem. Thus, for sufficiently large networks, finding global optima is generally easy, and they are in a sense ubiquitous.

## 3 The Hardness of Learning Neural Networks

We now review several known hardness results and apply them to our learning setting. For simplicity, throughout most of this section we focus on the PAC model in the binary classification case, over the Boolean cube, in the realizable case, and with a fixed target accuracy.[2]

Fix some $\epsilon, \delta \in (0, 1)$. For every dimension $d$, let the input space be $\mathcal{X}_d = \{0, 1\}^d$ and let $H$ be a hypothesis class of functions from $\mathcal{X}_d$ to $\{\pm 1\}$. We often omit the subscript $d$ when it is clear from context. A learning algorithm $A$ has access to an oracle that samples $\mathbf{x}$ according to an unknown distribution $D$ over $\mathcal{X}$ and returns $(\mathbf{x}, f^*(\mathbf{x}))$, where $f^*$ is some unknown target hypothesis in $H$. The objective of the algorithm is to return a classifier $f : \mathcal{X} \to \{\pm 1\}$, such that with probability of at least $1 - \delta$,

$$\mathbb{P}_{\mathbf{x} \sim D} \left[ f(\mathbf{x}) \neq f^*(\mathbf{x}) \right] \leq \epsilon.$$

We say that $A$ is efficient if it runs in time $\mathrm{poly}(d)$ and the function it returns can also be evaluated on a new instance in time $\mathrm{poly}(d)$. If there is such $A$, we say that $H$ is *efficiently learnable*.

In the context of neural networks, every network architecture defines a hypothesis class, $\mathcal{N}_{t,n,\sigma}$, that contains all target functions $f$ that can be implemented using a neural network with $t$ layers, $n$ neurons (excluding input neurons), and an activation function $\sigma$. The immediate question is which $\mathcal{N}_{t,n,\sigma}$ are efficiently learnable. We will first address this question for the threshold activation function, $\sigma_{0,1}(z) = 1$ if $z > 0$ and $0$ otherwise.

Observing that depth-2 networks with the threshold activation function can implement intersections of halfspaces, we will rely on the following hardness results, due to [15].

**Theorem 2** (Theorem 1.2 in [15]). *Let $\mathcal{X} = \{\pm 1\}^d$, let*

$$H^a = \left\{ \mathbf{x} \to \sigma_{0,1} \left( \mathbf{w}^\top \mathbf{x} - b - 1/2 \right) \ : \ b \in \mathbb{N}, \ \mathbf{w} \in \mathbb{N}^d, |b| + \|\mathbf{w}\|_1 \leq \mathrm{poly}(d) \right\},$$

*and let $H_k^a = \{ \mathbf{x} \to h_1(\mathbf{x}) \land h_2(\mathbf{x}) \land \ldots \land h_k(\mathbf{x}) : \forall i, h_i \in H^a \}$, where $k = d^\rho$ for some constant $\rho > 0$. Then under a certain cryptographic assumption, $H_k^a$ is not efficiently learnable.*

Under a different complexity assumption, [12] showed a similar result even for $k = \omega(1)$.

As mentioned before, neural networks of depth $\geq 2$ and with the $\sigma_{0,1}$ activation function can express intersections of halfspaces: For example, the first layer consists of $k$ neurons computing the $k$ halfspaces, and the second layer computes their conjunction by the mapping $\mathbf{x} \mapsto \sigma_{0,1} \left( \sum_i x_i - k + 1/2 \right)$. Trivially, if some class $H$ is not efficiently learnable, then any class containing it is also not efficiently learnable. We thus obtain the following corollary:

**Corollary 1.** *For every $t \geq 2, n = \omega(1)$, the class $\mathcal{N}_{t,n,\sigma_{0,1}}$ is not efficiently learnable (under the complexity assumption given in [12]).*

What happens when we change the activation function? In particular, two widely used activation functions for neural networks are the sigmoidal activation function, $\sigma_{\mathrm{sig}}(z) = 1/(1 + \exp(-z))$, and the rectified linear unit (ReLU) activation function, $\sigma_{\mathrm{relu}}(z) = \max\{z, 0\}$.

As a first observation, note that for $|z| \gg 1$ we have that $\sigma_{\mathrm{sig}}(z) \approx \sigma_{0,1}(z)$. Our data domain is the discrete Boolean cube, hence if we allow the weights of the network to be arbitrarily large, then $\mathcal{N}_{t,n,\sigma_{0,1}} \subseteq \mathcal{N}_{t,n,\sigma_{\mathrm{sig}}}$. Similarly, the function $\sigma_{\mathrm{relu}}(z) - \sigma_{\mathrm{relu}}(z-1)$ equals $\sigma_{0,1}(z)$ for every $|z| \geq 1$. As a result, without restricting the weights, we can simulate each threshold activated neuron by two ReLU activated neurons, which implies that $\mathcal{N}_{t,n,\sigma_{0,1}} \subseteq \mathcal{N}_{t,2n,\sigma_{\mathrm{relu}}}$. Hence, Corollary 1 applies to both sigmoidal networks and ReLU networks as well, as long as we do not regularize the weights of the network.

What happens when we do regularize the weights? Let $\mathcal{N}_{t,n,\sigma,L}$ be all target functions that can be implemented using a neural network of depth $t$, size $n$, activation function $\sigma$, and when we restrict the input weights of each neuron to be $\|\mathbf{w}\|_1 + |b| \leq L$.

One may argue that in many real world distributions, the difference between the two classes, $\mathcal{N}_{t,n,\sigma,L}$ and $\mathcal{N}_{t,n,\sigma_{0,1}}$ is small. Roughly speaking, when the distribution density is low around the decision boundary of neurons (similarly to separation with margin assumptions), then sigmoidal neurons will be able to effectively simulate threshold activated neurons.

In practice, the sigmoid and ReLU activation functions are advantageous over the threshold activation function, since they can be trained using gradient based methods. Can these empirical successes be turned into formal guarantees? Unfortunately, a closer examination of Thm. 2 demonstrates that if $L = \Omega(d)$ then learning $\mathcal{N}_{2,n,\sigma_{\mathrm{sig}},L}$ and $\mathcal{N}_{2,n,\sigma_{\mathrm{relu}},L}$ is still hard. Formally, to apply these networks to binary classification, we follow a standard definition of learning with a margin assumption: We assume that the learner receives examples of the form $(\mathbf{x}, \mathrm{sign}(f^*(\mathbf{x})))$ where $f^*$ is a real-valued function that comes from the hypothesis class, and we further assume that $|f^*(\mathbf{x})| \geq 1$. Even under this margin assumption, we have the following:

**Corollary 2.** *For every $t \geq 2$, $n = \omega(1)$, $L = \Omega(d)$, the classes $\mathcal{N}_{t,n,\sigma_{\mathrm{sig}},L}$ and $\mathcal{N}_{t,n,\sigma_{\mathrm{relu}},L}$ are not efficiently learnable (under the complexity assumption given in [12]).*

A proof is provided in the appendix. What happens when $L$ is much smaller? Later on in the paper we will show positive results for $L$ being a constant and the depth being fixed. These results will be obtained using polynomial networks, which we study in the next section.

## 4 Polynomial Networks

In the previous section we have shown several strong negative results for learning neural networks with the threshold, sigmoidal, and ReLU activation functions. One way to circumvent these hardness results is by considering another activation function. Maybe the simplest non-linear function is the squared function, $\sigma_2(x) = x^2$. We call networks that use this activation function *polynomial networks*, since they compute polynomial functions of their inputs. As in the previous section, we denote by $\mathcal{N}_{t,n,\sigma_2,L}$ the class of functions that can be implemented using a neural network of depth $t$, size $n$, squared activation function, and a bound $L$ on the $\ell_1$ norm of the input weights of each neuron. Whenever we do not specify $L$ we refer to polynomial networks with unbounded weights.

Below we study the expressiveness and computational complexity of polynomial networks. We note that algorithms for efficiently learning (real-valued) sparse or low-degree polynomials has been studied in several previous works (e.g. [13, 14, 8, 2, 1]). However, these rely on strong distributional assumptions, such as the data instances having a uniform or log-concave distribution, while we are interested in a distribution-free setting.

### 4.1 Expressiveness

We first show that, similarly to networks with threshold activation, polynomial networks of polynomial size can express all functions that can be implemented efficiently using a Turing machine.

**Theorem 3** (Polynomial networks can express Turing Machines). *Let $\mathcal{F}_d$ and $T$ be as in Thm. 1. Then there exist constants $b, c \in \mathbb{R}_+$ such that for every $d$, the class $\mathcal{N}_{t,n,\sigma_2,L}$, with $t = c\,T(d)\log(T(d)) + b$, $n = t^2$, and $L = b$, contains $\mathcal{F}_d$.*

The proof of the theorem relies on the result of [18] and is given in the appendix.

Another relevant expressiveness result, which we will use later, shows that polynomial networks can approximate networks with sigmoidal activation functions:

**Theorem 4.** *Fix $0 < \epsilon < 1$, $L \geq 3$ and $t \in \mathbb{N}$. There are $B_t \in \tilde{O}(\log(tL + L\log\frac{1}{\epsilon}))$ and $B_n \in \tilde{O}(tL + L\log\frac{1}{\epsilon})$ such that for every $f \in \mathcal{N}_{t,n,\sigma_{\mathrm{sig}},L}$ there is a function $g \in \mathcal{N}_{tB_t,nB_n,\sigma_2}$, such that $\sup_{\|\mathbf{x}\|_\infty < 1} \|f(\mathbf{x}) - g(\mathbf{x})\|_\infty \leq \epsilon$.*

The proof relies on an approximation of the sigmoid function based on Chebyshev polynomials, as was done in [21], and is given in the appendix.

## 4.2 Training Time

We now turn to the computational complexity of learning polynomial networks. We first show that it is hard to learn polynomial networks of depth $\Omega(\log(d))$. Indeed, by combining Thm. 4 and Corollary 2 we obtain the following:

**Corollary 3.** *The class $\mathcal{N}_{t,n,\sigma_2}$, where $t = \Omega(\log(d))$ and $n = \Omega(d)$, is not efficiently learnable.*

On the flip side, constant-depth polynomial networks can be learned in polynomial time, using a simple linearization trick. Specifically, the class of polynomial networks of constant depth $t$ is contained in the class of multivariate polynomials of total degree at most $s = 2^t$. This class can be represented as a $d^s$-dimensional linear space, where each vector is the coefficient vector of some such polynomial. Therefore, the class of polynomial networks of depth $t$ can be learned in time $\text{poly}(d^{2^t})$, by mapping each instance vector $\mathbf{x} \in \mathbb{R}^d$ to all of its monomials, and learning a linear predictor on top of this representation (which can be done efficiently in the realizable case, or when a convex loss function is used). In particular, if $t$ is a constant then so is $2^t$ and therefore polynomial networks of constant depth are efficiently learnable. Another way to learn this class is using support vector machines with polynomial kernels.

An interesting application of this observation is that depth-2 sigmoidal networks are efficiently learnable with sufficient regularization, as formalized in the result below. This contrasts with corollary 2, which provides a hardness result without regularization.

**Theorem 5.** *The class $\mathcal{N}_{2,n,\sigma_{\text{sig}},L}$ can be learned, to accuracy $\epsilon$, in time $\text{poly}(T)$ where $T = (1/\epsilon) \cdot O(d^{4L \ln(11L^2+1)})$.*

The idea of the proof is as follows. Suppose that we obtain data from some $f \in \mathcal{N}_{2,n,\sigma_{\text{sig}},L}$. Based on Thm. 4, there is $g \in \mathcal{N}_{2B_t,nB_n,\sigma_2}$ that approximates $f$ to some fixed accuracy $\epsilon_0 = 0.5$, where $B_t$ and $B_n$ are as defined in Thm. 4 for $t = 2$. Now we can learn $\mathcal{N}_{2B_t,nB_n,\sigma_2}$ by considering the class of all polynomials of total degree $2^{2B_t}$, and applying the linearization technique discussed above. Since $f$ is assumed to separate the data with margin 1 (i.e. $y = \text{sign}(f^*(\mathbf{x})), |f^*(\mathbf{x})| \geq 1$), then $g$ separates the data with margin 0.5, which is enough for establishing accuracy $\epsilon$ in sample and time that depends polynomially on $1/\epsilon$.

## 4.3 Learning 2-layer and 3-layer Polynomial Networks

While interesting theoretically, the above results are not very practical, since the time and sample complexity grow very fast with the depth of the network.[3] In this section we describe practical, provably correct, algorithms for the special case of depth-2 and depth-3 polynomial networks, with some additional constraints. Although such networks can be learned in polynomial time via explicit linearization (as described in section 4.2), the runtime and resulting network size scales quadratically (for depth-2) or cubically (for depth-3) with the data dimension $d$. In contrast, our algorithms and guarantees have a much milder dependence on $d$.

We first consider 2 layer polynomial networks, of the following form:

$$\mathcal{P}_{2,k} = \left\{ \mathbf{x} \mapsto b + \mathbf{w}_0^\top \mathbf{x} + \sum_{i=1}^{k} \alpha_i (\mathbf{w}_i^\top \mathbf{x})^2 : \; \forall i \geq 1, |\alpha_i| \leq 1, \|\mathbf{w}_i\|_2 = 1 \right\} .$$

This networks corresponds to one hidden layer containing $r$ neurons with the squared activation function, where we restrict the input weights of all neurons in the network to have bounded $\ell_2$ norm, and where we also allow a direct linear dependency between the input layer and the output layer.

We'll describe an efficient algorithm for learning this class, which is based on the GECO algorithm for convex optimization with low-rank constraints [20].

The goal of the algorithm is to find $f$ that minimizes the objective

$$R(f) = \frac{1}{m} \sum_{i=1}^{m} \ell(f(\mathbf{x}_i), y_i), \tag{1}$$

where $\ell : \mathbb{R} \times \mathbb{R} \to \mathbb{R}$ is a loss function. We'll assume that $\ell$ is $\beta$-smooth and convex.

The basic idea of the algorithm is to gradually add hidden neurons to the hidden layer, in a greedy manner, so as to decrease the loss function over the data. To do so, define $\mathcal{V} = \{\mathbf{x} \mapsto (\mathbf{w}^\top \mathbf{x})^2 : \|\mathbf{w}\|_2 = 1\}$ the set of functions that can be implemented by hidden neurons. Then every $f \in \mathcal{P}_{2,r}$ is an affine function plus a weighted sum of functions from $\mathcal{V}$. The algorithm starts with $f$ being the minimizer of $R$ over all affine functions. Then at each greedy step, we search for $g \in \mathcal{V}$ that minimizes a first order approximation of $R(f + \eta g)$:

$$R(f + \eta g) \approx R(f) + \eta \frac{1}{m} \sum_{i=1}^{m} \ell'(f(\mathbf{x}_i), y_i) g(\mathbf{x}_i) , \tag{2}$$

where $\ell'$ is the derivative of $\ell$ w.r.t. its first argument. Observe that for every $g \in \mathcal{V}$ there is some $\mathbf{w}$ with $\|\mathbf{w}\|_2 = 1$ for which $g(\mathbf{x}) = (\mathbf{w}^\top \mathbf{x})^2 = \mathbf{w}^\top \mathbf{x}\mathbf{x}^\top \mathbf{w}$. Hence, the right-hand side of Eq. (2) can be rewritten as $R(f) + \eta\, \mathbf{w}^\top \left( \frac{1}{m} \sum_{i=1}^{m} \ell'(f(\mathbf{x}_i), y_i) \mathbf{x}_i \mathbf{x}_i^\top \right) \mathbf{w}$ . The vector $\mathbf{w}$ that minimizes this expression (for positive $\eta$) is the leading eigenvector of the matrix $\left( \frac{1}{m} \sum_{i=1}^{m} \ell'(f(\mathbf{x}_i), y_i) \mathbf{x}_i \mathbf{x}_i^\top \right)$. We add this vector as a hidden neuron to the network.[4] Finally, we minimize $R$ w.r.t. the weights from the hidden layer to the output layer (namely, w.r.t. the weights $\alpha_i$).

The following theorem, which follows directly from Theorem 1 of [20], provides convergence guarantee for GECO. Observe that the theorem gives guarantee for learning $\mathcal{P}_{2,k}$ if we allow to output an over-specified network.

**Theorem 6.** *Fix some $\epsilon > 0$. Assume that the loss function is convex and $\beta$-smooth. Then if the GECO Algorithm is run for $r > \frac{2\beta k^2}{\epsilon}$ iterations, it outputs a network $f \in \mathcal{N}_{2,r,\sigma_2}$ for which $R(f) \le \min_{f^* \in \mathcal{P}_{2,k}} R(f^*) + \epsilon$.*

We next consider a hypothesis class consisting of third degree polynomials, which is a subset of 3-layer polynomial networks (see Lemma 1 in the appendix) . The hidden neurons will be functions from the class: $\mathcal{V} = \cup_{i=1}^{3} \mathcal{V}_i$ where $\mathcal{V}_i = \left\{ \mathbf{x} \mapsto \prod_{j=1}^{i} (\mathbf{w}_j^\top \mathbf{x}) : \forall j, \|\mathbf{w}_j\|_2 = 1 \right\}$ . The hypothesis class we consider is $\mathcal{P}_{3,k} = \left\{ \mathbf{x} \mapsto \sum_{i=1}^{k} \alpha_i g_i(\mathbf{x}) : \forall i, |\alpha_i| \le 1, g_i \in \mathcal{V} \right\}$.

The basic idea of the algorithm is the same as for 2-layer networks. However, while in the 2-layer case we could implement efficiently each greedy step by solving an eigenvalue problem, we now face the following tensor approximation problem at each greedy step:

$$\max_{g \in \mathcal{V}_3} \frac{1}{m} \sum_{i=1}^{m} \ell'(f(\mathbf{x}_i), y_i) g(\mathbf{x}_i) = \max_{\|\mathbf{w}\|=1, \|\mathbf{u}\|=1, \|\mathbf{v}\|=1} \frac{1}{m} \sum_{i=1}^{m} \ell'(f(\mathbf{x}_i), y_i)(\mathbf{w}^\top \mathbf{x}_i)(\mathbf{u}^\top \mathbf{x}_i)(\mathbf{v}^\top \mathbf{x}_i) .$$

While this is in general a hard optimization problem, we can approximate it – and luckily, an approximate greedy step suffices for success of the greedy procedure. This procedure is given in Figure 1, and is again based on an approximate eigenvector computation. A guarantee for the quality of approximation is given in the appendix, and this leads to the following theorem, whose proof is given in the appendix.

**Theorem 7.** *Fix some $\delta, \epsilon > 0$. Assume that the loss function is convex and $\beta$-smooth. Then if the GECO Algorithm is run for $r > \frac{4d\beta k^2}{\epsilon(1-\tau)^2}$ iterations, where each iteration relies on the approximation procedure given in Fig. 1, then with probability $(1-\delta)^r$, it outputs a network $f \in \mathcal{N}_{3,5r,\sigma_2}$ for which $R(f) \le \min_{f^* \in \mathcal{P}_{3,k}} R(f^*) + \epsilon$.*

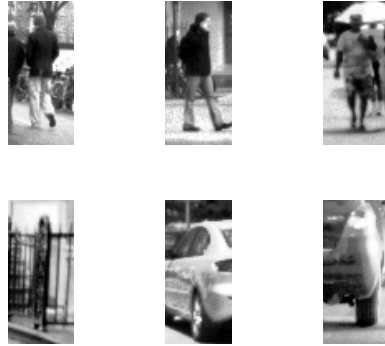

$$\begin{array}{|l|}
\hline
\textbf{Input}: \{x_i\}_{i=1}^m \in \mathbb{R}^d \; \alpha \in \mathbb{R}^m, \tau, \delta \\
\textbf{Output}: \text{A } \frac{1-\tau}{\sqrt{d}} \text{ approximate solution to} \\
\displaystyle\max_{\|\mathbf{w}\|,\|\mathbf{u}\|,\|\mathbf{v}\|=1} F(\mathbf{w},\mathbf{u},\mathbf{v}) = \sum_i \alpha_i (\mathbf{w}^\top \mathbf{x}_i)(\mathbf{u}^\top \mathbf{x}_i)(\mathbf{v}^\top \mathbf{x}_i) \\
\text{Pick randomly } \mathbf{w}_1,\ldots,\mathbf{w}_s \text{ iid according to } \mathcal{N}(0, I_d). \\
\textbf{For } t = 1,\ldots,2d\log\frac{1}{\delta} \\
\quad \mathbf{w}_t \leftarrow \frac{\mathbf{w}_t}{\|\mathbf{w}_t\|} \\
\quad \text{Let } A = \sum_i \alpha_i (\mathbf{w}_t^\top \mathbf{x}_i)\mathbf{x}_i\mathbf{x}_i^\top \text{ and set } \mathbf{u}_t, \mathbf{v}_t \text{ s.t:} \\
\quad Tr(\mathbf{u}_t^\top A \mathbf{v}_t) \geq (1-\tau)\max_{\|\mathbf{u}\|,\|\mathbf{v}\|=1} Tr(\mathbf{u}^\top A \mathbf{v}). \\
\text{Return } \mathbf{w}, \mathbf{u}, \mathbf{v} \text{ the maximizers of } \max_{i \leq s} F(\mathbf{w}_i, \mathbf{u}_i, \mathbf{u}_i). \\
\hline
\end{array}$$

Figure 1: Approximate tensor maximization.

# 5 Experiments

To demonstrate the practicality of GECO to train neural networks for real world problems, we considered a pedestrian detection problem as follows. We collected 200k training examples of image patches of size 88x40 pixels containing either pedestrians (positive examples) or hard negative examples (containing images that were classified as pedestrians by applying a simple linear classifier in a sliding window manner). See a few examples of images above. We used half of the examples as a training set and the other half as a test set. We calculated HoG features ([11]) from the images[5]. We then trained, using GECO, a depth-2 polynomial network on the resulting features. We used 40 neurons in the hidden layer. For comparison we trained the same network architecture (i.e. 40 hidden neurons with a squared activation function) by SGD. We also trained a similar network (40 hidden neurons again) with the ReLU activation function. For the SGD implementation we tried the following tricks to speed up the convergence: heuristics for initialization of the weights, learning rate rules, mini-batches, Nesterov's momentum (as explained in [23]), and dropout. The test errors of SGD as a function of the number of iterations are depicted on the top plot of the Figure on the side. We also mark the performance of GECO as a straight line (since it doesn't involve SGD iterations). As can be seen, the error of GECO is slightly better than SGD. It should be also noted that we had to perform a very large number of SGD iterations to obtain a good solution, while the runtime of GECO was much faster. This indicates that GECO may be a valid alternative approach to SGD for training depth-2 networks. It is also apparent that the squared activation function is slightly better than the ReLU function for this task.

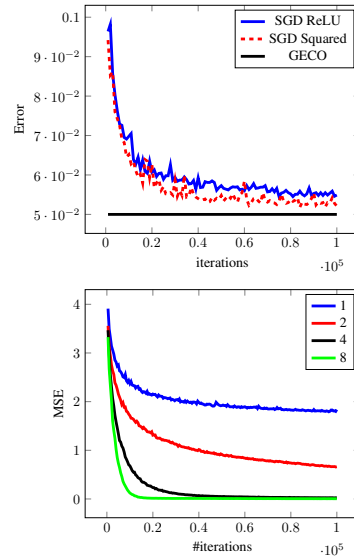

The second plot of the side figure demonstrates the benefit of over-specification for SGD. We generated random examples in $\mathbb{R}^{150}$ and passed them through a random depth-2 network that contains 60 hidden neurons with the ReLU activation function. We then tried to fit a new network to this data with over-specification factors of $1, 2, 4, 8$ (e.g., over-specification factor of 4 means that we used $60 \cdot 4 = 240$ hidden neurons). As can be clearly seen, SGD converges much faster when we over-specify the network.

**Acknowledgements:** This research is supported by Intel (ICRI-CI). OS was also supported by an ISF grant (No. 425/13), and a Marie-Curie Career Integration Grant. SSS and RL were also supported by the MOS center of Knowledge for AI and ML (No. 3-9243). RL is a recipient of the Google Europe Fellowship in Learning Theory, and this research is supported in part by this Google Fellowship. We thank Itay Safran for spotting a mistake in a previous version of Sec. 2 and to James Martens for helpful discussions.

## Footnotes

[1]For example, consider the function computed by the first layer, $X \mapsto \sigma(V_d X)$, where $\sigma$ is a sigmoid function. Since $\sigma$ is non-linear, the columns of $\sigma(V_d X)$ will not be linearly dependent in general.

[2]While we focus on the realizable case (i.e., there exists $f^* \in H$ that provides perfect predictions), with a fixed accuracy ($\epsilon$) and confidence ($\delta$), since we are dealing with hardness results, the results trivially apply to the agnostic case and to learning with arbitrarily small accuracy and confidence parameters.

[3]If one uses SVM with polynomial kernels, the time and sample complexity may be small under margin assumptions in a feature space corresponding to a given kernel. Note, however, that large margin in that space is very different than the assumption we make here, namely, that there is a network with a small number of hidden neurons that works well on the data.

[4]It is also possible to find an approximate solution to the eigenvalue problem and still retain the performance guarantees (see [20]). Since an approximate eigenvalue can be found in time $O(d)$ using the power method, we obtain the runtime of GECO depends linearly on $d$.

[5]Using the Matlab implementation provided in `http://www.mathworks.com/matlabcentral/fileexchange/33863-histograms-of-oriented-gradients`.

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
