[Supplementary Material]

# On the Computational Efficiency of Training Neural Networks – Appendix

**Roi Livni**
The Hebrew University
roi.livni@mail.huji.ac.il

**Shai Shalev-Shwartz**
The Hebrew University
shais@cs.huji.ac.il

**Ohad Shamir**
Weizmann Institute
ohad.shamir@weizmann.ac.il

## A  Proofs

### A.1  Proof of Corollary 2

#### A.1.1  Hardness result for the class $\mathcal{N}_{2,n,\sigma_{\mathrm{sig}},L}$:

Consider $H^a$ as defined in Thm. 2. Note that for every $h \in H^a$ there are integral $\mathbf{w}$ and $b$ such that $h(\mathbf{x}) = \mathbf{w}^\top \mathbf{x} - b - \frac{1}{2}$ and we have that $|h(\mathbf{x})| \geq 1/2$. Given $k$ hyperplanes $\{h_i\}_{i=1}^k$ consider the neurons

$$g_i(\mathbf{x}) = 1/\left(1 + \exp\left(-Ch_i(\mathbf{x})\right)\right),$$

where $C \in \omega(1)$ is to be chosen later. Let

$$g(\mathbf{x}) = \frac{Cd}{2k + \frac{1}{3}}\left(\sum_{i=1}^{k} g_i(\mathbf{x}) - k + \frac{1}{3}\right).$$

If $\|\mathbf{w}\|_1 + |b + \frac{1}{2}| \leq d$ we have that $g(\mathbf{x}) \in \mathcal{N}_{2,n,\sigma_{\mathrm{sig}},L}$, whenever $L \geq Cd$. Choose $C \in O(k)$ sufficiently large so that

$$1/\left(1 + \exp\left(-\frac{C}{2}\right)\right) - 1 > -\frac{1}{3k}.$$

and

$$1/\left(1 + \exp\left(\frac{C}{2}\right)\right) < \frac{2}{3}.$$

Since $|h_i(\mathbf{x})| \geq \frac{1}{2}$ for all $i$, if the output of all neurons $\{g_i\}$ is positive we have

$$\frac{2k + 1/3}{Cd} g(\mathbf{x}) \geq k\left(\frac{1}{1 + \exp(-\frac{C}{2})} - 1\right) + \frac{1}{3} > 0.$$

On the other hand, if $h_i(\mathbf{x}) < -\frac{1}{2}$ for some $i$ we have that

$$\frac{2k + 1/3}{Cd} g(x) \leq \sum_{i=1}^{k-1} g_i(\mathbf{x}) + \frac{1}{1 + \exp(\frac{C}{2})} - k + \frac{1}{3} \leq k - 1 - k + \frac{1}{1 + \exp\frac{C}{2}} + \frac{1}{3} < 0.$$

We've demonstrated that the target function $\mathrm{sign}(g(\mathbf{x}))$ implements $h_1 \wedge h_2 \wedge \ldots \wedge h_k$ thus $H_k^a \subseteq \mathcal{N}_{2,k+1,\sigma_{\mathrm{sig}},L}$.

#### A.1.2  Hardness result for the class $\mathcal{N}_{2,n,\sigma_{\mathrm{relu}},L}$:

Again, given $k$ hyperplanes $\{h_i\}_{i=1}^k$, for every $k$ consider the two neurons:

$$g_i^+(\mathbf{x}) = \max\{0, 2h_i(\mathbf{x})\}, \quad g_i^- = (\mathbf{x})\max\{0, 2h_i(\mathbf{x}) - 1\}.$$

And let

$$g(\mathbf{x}) = \frac{1}{2k}\left(\sum_{i=1}^{k}\left(g_i^+(\mathbf{x}) - g_i^-(\mathbf{x})\right) - k\right).$$

As before $g(\mathbf{x}) \in \mathcal{N}_{2,2k+1,\sigma_{\mathrm{relu}},L}$, whenever $L \geq 2d$. One can also verify that $g(\mathbf{x})$ implements $h_1 \wedge h_2 \wedge \ldots \wedge h_k$.

## A.2  Proof of Thm. 3

We start by showing that we can implement AND, OR, NEG, Id gates using polynomial networks of fixed depth and size. As a corollary, we can implement circuits with fixed number of fan-ins. NEG can be implemented with $x \mapsto 1-x$ and Id can be implemented with $x \mapsto \frac{1}{4}\left((x+1)^2 - (x-1)^2\right)$. Next note that

$$\mathrm{AND}(\mathbf{x}_1, \mathbf{x}_2) = \mathbf{x}_1 \cdot \mathbf{x}_2, \quad \text{and} \quad \mathrm{OR}(\mathbf{x}_1, \mathbf{x}_2) = \mathbf{x}_1 + \mathbf{x}_2 - \mathrm{AND}(\mathbf{x}_1, \mathbf{x}_2).$$

and that $\mathbf{x}_1 \cdot \mathbf{x}_2 = \frac{1}{4}\left((\mathbf{x}_2 + \mathbf{x}_1)^2 - (\mathbf{x}_2 - \mathbf{x}_1)^2\right)$. Thus we can implement with two layers a conjunction and disjunction of 2 neurons. By adding a fixed number of layers, we can also implement the conjunction and disjunction of any fixed number of neurons. Therefore, if $B$ is a circuit with fixed number of fan-ins, of size T, we can implement it using a polynomial network with $O(T)$ layers and $O(T^2)$ neurons, where layer $t$ simulates the calculation of all gates at depth $t$.

Now, by [1], any Turing machine with runtime $T$ can be simulated by an *oblivious* Turing machine with $O(T \log T)$-steps. An oblivious Turing machine is a machine such that the position of the machine head at time $t$ does not depend on the input of the machine (and therefore is known ahead of time). We can now easily simulate the machine by a network of depth $O(T \log T)$, where the nodes at each layer contain the state of the turing machine (the content of the tape and the position at the state machine), and the transition from layer to layer depends on a constant size circuit, and hence can be implemented by a constant depth polynomial network.

## A.3  Proof of Thm. 4

The idea of proof of Thm. 4 is as follows: First we show that we can express any $T$-degree polynomial using $O(\log T)$ layers and $O(T)$ neurons. This is done in Lemma 1 part 4. As a second step, we show in Lemma 2 that a sigmoidal function can be approximated in a ball of radius $L$ by a $O(\log \frac{L}{\epsilon})$-degree polynomial. The result follows by replacing each sigmoid activation unit with added layers that approximate the sigmoidal function on the output of the previous layer. We will first prove the two Lemmas. The proof of Thm. 4 is then given at the end of the section.

**Lemma 1.** *The following statements hold:*

1. *If $g \in \mathcal{N}_{t,n,\sigma_2,L}$ for some $L \geq 2$ then $g \in \mathcal{N}_{t',n+2(t'-t),\sigma_2,L}$ for every $t' \geq t$.*

2. *If $G \in \mathcal{N}_{t,n,\sigma_2,L}$ for some $L \geq 2$ and $G$ is a network with two output neurons $g_1$ and $g_2$ then $g_1 \cdot g_2 \in \mathcal{N}_{t+1,n+1,\sigma_2}$.*

3. *If $g \in \mathcal{N}_{t,n,\sigma_2,L}$ for some $L \geq 2$ then $(g)^T \in \mathcal{N}_{t',n',\sigma_2,L}$. where $t' = t + \log T + \log \log T$ and $n' = n + 2\log T + \log T(\log \log T)$.*

4. *If $g \in \mathcal{N}_{t,n,\sigma_2,L}$ then $\sum_{i=1}^{T} a_i(g(\mathbf{x}))^k$ is in $\mathcal{N}_{t',n',\sigma_2,L'}$ where*

    $$t' = t + \log T + \log \log T, \quad n' = n + 2\|a\|_0(2\log T + \log T(\log \log T)),$$

    *where $\|a\|_0 = |\{k : a_k \neq 0\}|$. And $L' = \max\{\|a\|_1, L, 2\}$.*

*Proof.*     1. Proof of 1]: Note that $\frac{1}{4}((x+1)^2 - (x-1)^2) = x$. Next we prove the statement by induction. For $t' = t$, the satement is trivial. Next assume that $g \in \mathcal{N}_{t,n+2(t'-t),\sigma_2,L}$. Let

    $$h_1(\mathbf{x}) = \left(\frac{1}{2}g(\mathbf{x}) + \frac{1}{2}\right)^2, \quad h_2(\mathbf{x}) = \left(\frac{1}{2}g(\mathbf{x}) - \frac{1}{2}\right)^2.$$

Let $h(\mathbf{x}) = h_1(\mathbf{x}) - h_2(\mathbf{x})$ then $h(\mathbf{x}) = g(\mathbf{x})$. By taking the network that implements $g$, removing the output neuron, adding an additional hidden layer that consists of $h_1$ and $h_2$ and finally adding an additional output neuron we have that $h(\mathbf{x}) \in \mathcal{N}_{t'+1,n+2(t'-t)+2,\sigma_2,L}$.

2. Proof of 2: Like before, note that $x_1 \cdot x_2 = \frac{1}{4}(x_1 + x_2)^2 - \frac{1}{4}(x_1 - x_2)^2$. Let

$$h_1(\mathbf{x}) = (\frac{1}{2}g_1(\mathbf{x}) + \frac{1}{2}g_2(\mathbf{x}))^2, \quad h_2(\mathbf{x}) = (\frac{1}{2}g_1(\mathbf{x}) + \frac{1}{2}g_2(\mathbf{x}))^2.$$

As before we remove from the network that implements $G$ the two nodes at the output layer, add an additional hidden layer that implements $h_1$ and $h_2$ and finally add an output neuron $h(\mathbf{x}) = h_1(\mathbf{x}) - h_2(\mathbf{x})$.

3. Proof of 3:Write $T = \sum_{i=1}^{\log T} \epsilon_i 2^i$ where $\epsilon_i = \{0, 1\}$.

We will first show that we can construct a polynomial network that contains in layer $t + \log T$ neurons $h_1, \ldots, h_{\log T}$ such that $h_k(\mathbf{x}) = (g(\mathbf{x}))^{2^k}$. It is easy to see that we can implement a neuron $h'(\mathbf{x})_k$ at layer $t + k$ such that $h'_k(\mathbf{x}) = (g(\mathbf{x}))^{2^k}$. Next, using 1 we add $2(\log T - k)$ neurons and implement $h'_k$ in layer $t + \log T$.

Finally, we implement $\prod_{\{i:\epsilon_i \neq 0\}} h_i(\mathbf{x})$ using $\log \log T$ layers and $\log T \log \log T$ additional neurons, this can be done by applying 2 where at each layer we pair the neurons at previous layer and do their product (e.g. if for every $i$ $\epsilon_i \neq 0$ then at the next layer we implement $(h_1 \cdot h_2, h_3 \cdot h_4, \ldots h_{t-1}h_t)$ then at the next layer we implement $(h_1 \cdot h_2 \cdot h_3 \cdot h_4, \ldots, h_{t-4} \cdots h_t)$ etc..)

4. Proof of 4: Follows from 1 and 3.

$\square$

**Lemma 2** (Sigmoidals are approximable via polynomial networks). *The following holds for any $\epsilon \geq 0$ and (for simplicity) $L \geq 3$: Set*

$$T = \log\left(2L^4 + \exp\left(7L \ln\left(\frac{4L}{\epsilon} + 3\right)\right)\right) + 2\log\frac{8}{\epsilon}.$$

*There is a polynomial $p(x) = \sum_{j=1}^{T} a_j x^j$, such that:*

$$\sup_{|x|<4L} |p(x) - \sigma_{\text{sig}}(x)| < \epsilon.$$

*Proof of Lemma 2.* Set

$$t' = \log\left(2L^4 + \exp(7L \ln\left(\frac{4L}{\epsilon} + 3\right)\right).$$

According to [3] Lemma 2, there is an analytic function $q$ such that

$$\sup_{|x|\leq 1} |q(x) - \sigma_{\text{sig}}(4Lx)| \leq \frac{\epsilon}{2},$$

and

$$q(x) = \sum_{j=0}^{\infty} \beta_j x^j$$

where

$$\sum_{j=0}^{\infty} \beta_j^2 2^j \leq 2^{t'}.$$

Note that for every $j$ we have $|\beta_j| \leq 2^{\frac{t'-j}{2}}$. Thus

$$\sup_{|x|<1} |\sum_{j>T} \beta_j x^j| \leq \sum_{j>T} |\beta_j| \leq \sum_{j>T} 2^{\frac{t'-j}{2}} = 2^{\frac{t'-T}{2}} \sum_{j=1}^{\infty} (\sqrt{2})^{-j} < 4 \cdot 2^{\frac{t'-T}{2}}.$$

Recalling that $T = t' + 2\log\frac{8}{\epsilon}$ and letting $p_0(x) = \sum_{j=0}^{T}\beta_j x^j$, we have by triangular inequality that

$$\sup_{|x|\leq 1}|p_0(x) - \sigma_{\text{sig}}(4Lx)| \leq \epsilon.$$

Finally, take $p(x) = p_0(\frac{x}{4L})$. $\qquad\square$

### A.3.1 Back to proof of Thm. 4

Set

$$T = \log\left(2L^4 + \exp(7L\ln\left(\frac{(4L)^t}{\epsilon} + 3\right))\right) + 2\log\frac{8(4L)^{t-1}}{\epsilon}.$$

and have

$$B_t = 1 + \log T + \log\log T \in \tilde{O}\left(\log L \log\frac{L^t}{\epsilon}\right), \tag{1}$$

$$B_n = 1 + 2T\left(2\log T + \log T \log\log T\right) \in \tilde{O}(L\log\frac{L^t}{\epsilon}). \tag{2}$$

We prove the statement by induction on $t$, our induction hypothesis will hold for networks with not necessarily a single output neuron. For $t = 1$, since $\mathcal{N}_{1,n,\sigma_2} = \mathcal{N}_{1,n,\sigma_{\text{sig}}}$, the statement is trivial. Next let $F \in \mathcal{N}_{t,n,\sigma_{\text{sig}},L}$, assume $F : \mathbb{R}^d \to \mathbb{R}^s$ (i.e. the output layer has $s$ nodes). There is a target function $F^{(t-1)} \in \mathcal{N}_{t-1,n-s,\sigma_{\text{sig}},L}$ such that for every $i = 1\ldots s$ we have

$$F_i = \mathbf{w}_i^\top \sigma_{\text{sig}}(F^{(t-1)}(\mathbf{x})).$$

where $\sigma_{\text{sig}}(F^{(t-1)}(\mathbf{x}))$ denotes pointwise activation of $\sigma_{\text{sig}}$ on the coordinates of $F^{(t-1)}$.

By induction, there is some $P^{(t-1)} \in \mathcal{N}_{(t-1)B_t,(n-s)B_n,\sigma_2}$ such that

$$\sup_{\|\mathbf{x}\|_\infty\leq 1}\|P^{(t-1)}(\mathbf{x}) - F^{(t-1)}(\mathbf{x})\|_\infty \leq \frac{\epsilon}{4L} \leq \frac{\epsilon}{4}$$

By Lemma 1 part 4 and Lemma 2 we can add $B_t$ layers and $B_n$ neurons and implement a new target function $P_i$ such that

$$P_i(\mathbf{x}) = \mathbf{w}_i^\top p(P^{(t-1)}(\mathbf{x})),$$

where $p$ is taken from Lemma 2 and satisfies

$$\sup_{|x|\leq 4L}|p(x) - \sigma_{\text{sig}}(x)| < \frac{\epsilon}{(4L)^{t-1}} \leq \frac{\epsilon}{2L},$$

Taken together we can add $sB_n$ nodes to implement a target function $P = P_1, \ldots, P_s$. Next,

$$\|P(\mathbf{x})-F(\mathbf{x})\|_\infty \leq \sup_i\|P_i(\mathbf{x})-\mathbf{w}_i^\top\sigma_{\text{sig}}(P^{(t-1)}(\mathbf{x}))\|+\|\mathbf{w}_i^\top\sigma_{\text{sig}}(P^{(t-1)}(\mathbf{x}))-\mathbf{w}_i^\top\sigma_{\text{sig}}(F^{(t-1)}(\mathbf{x}))\|.$$

Recall that the $\ell_1$-norm of each weight vector of each neuron is bounded by $L$ and that the output of each neuron is bounded by $\sup_x \sigma_{\text{sig}}(x) = 1$, hence: $\|F^{(t-1)}(\mathbf{x})\|_\infty \leq L$. By induction we also have that $\|F^{(t-1)}(\mathbf{x}) - P^{(t-1)}(\mathbf{x})\|_\infty \leq 1$ hence $\|P^{t-1}(\mathbf{x})\|_\infty \leq 2L$ and we have:

$$\sup_i\|\mathbf{w}_i^\top p(P^{(t-1)}(\mathbf{x}))-\mathbf{w}_i^\top\sigma_{\text{sig}}(P^{(t-1)}(\mathbf{x}))\|+\sup_i\|\mathbf{w}_i^\top\sigma_{\text{sig}}(P^{(t-1)}(\mathbf{x}))-\mathbf{w}_i^\top\sigma_{\text{sig}}(F^{(t-1)}(\mathbf{x}))\| \leq .$$

$$\frac{\|\mathbf{w}_i\|_1\epsilon}{2L} + \|\mathbf{w}_i\|_1\|P^{(t-1)}(\mathbf{x}) - F^{(t-1)}(\mathbf{x})\|_\infty \leq \frac{\epsilon}{2} + \frac{\epsilon}{4} \leq \epsilon.$$

Where we used the fact that $\sigma_{\text{sig}}$ is 1-Lipschitz.

## A.4 Proof of Thm. 7

That $f \in \mathcal{N}_{3,5r,\sigma_2}$ can be shown using Lemma 1 and the output's structure.

Let us denote by $\mathrm{Approx}(\frac{(1-\tau)}{\sqrt{2d}}, \nabla R(f))$, a procedure that returns $g \in \mathcal{V}$ such that

$$\sum_{i=1}^{m} \ell'(f(\mathbf{x}_i), y_i) g(\mathbf{x}_i) \geq \frac{(1-\tau)}{\sqrt{2d}} \max_{g^* \in \mathcal{V}} \frac{1}{m} \sum_{i=1}^{m} \ell'(f(\mathbf{x}_i), y_i) g^*(\mathbf{x}_i)$$

---

Input: $r$ $\tau$, $\epsilon$
Initialize: $\mathcal{W} = \emptyset$, $f = 0$
**For** $t = 1, \ldots, r$

   Set $g(\mathbf{x}) := \mathrm{Approx}\left(\frac{(1-\tau)}{\sqrt{2d}}, \nabla R(f)\right)$
   Add $g(\mathbf{x})$ to $\mathcal{W}$.
   Let $\alpha^{(t)}$ and $f^{(t)}$ optimize the problem $\min_f \frac{1}{m} \sum_{i=1}^{m} \ell(f(\mathbf{x}_i), y_i)$
   subject to $f(\mathbf{x}) = \sum_{g \in \mathcal{W}} \alpha_g g(\mathbf{x})$.

Return: $f = f^{(r)}$.

---

Figure 3: GECO with different $\mathrm{Approx}$ procedure.

The GECO algorithm is presented in Fig. 3 with an $\mathrm{Approx}$ procedure that is implemented with respect to $\mathcal{V} = \cup \mathcal{V}_i$. The guarantees in Thm. 7 are proven in exactly the same manner as in [2].

The remained challenge is to demonstrate that the $\mathrm{Approx}$ procedure can be implemented efficiently, relying on the algorithm presented in Fig. 1 . To this end, note that the only difficulty is when $g^* \in \mathcal{V}_3$ (since if $g^* \in \mathcal{V}_2$ or $g^* \in \mathcal{V}_1$ we are back to the 2-layer scenario). The proof follows directly from the following lemma:

**Lemma 3.** *Let $\mathbf{w}^*, \mathbf{u}^*, \mathbf{v}^*$ be the output of the Algorithm presented in Fig. 1 with parameters $\delta, \tau, \{\mathbf{x}_i\}_{i=1}^{m}$ and $\alpha_i = \ell'(f(\mathbf{x}_i), y_i)$. With probability at least $1 - \delta$:*

$$F(\mathbf{w}^*, \mathbf{u}^*, \mathbf{v}^*) \geq \frac{1-\tau}{\sqrt{2d}} \max_{\|\mathbf{w}\|, \|\mathbf{u}\|, \|\mathbf{v}\| \leq 1} F(\mathbf{w}, \mathbf{u}, \mathbf{v}),$$

*where*

$$F(\mathbf{w}, \mathbf{u}, \mathbf{v}) = \frac{1}{m} \sum_{i=1}^{m} \ell'(f(\mathbf{x}_i), y_i)(\mathbf{w}^\top \mathbf{x}_i) \cdot (\mathbf{u}^\top \mathbf{x}_i) \cdot (\mathbf{v}^\top \mathbf{x}_i).$$

*Proof.* Let us denote by $\mathbf{w}^*, \mathbf{u}^*, \mathbf{v}^*$ the maximizers of $F(\mathbf{w}, \mathbf{u}, \mathbf{v})$, over all $\|\mathbf{w}\|, \|\mathbf{u}\|, \|\mathbf{v}\| = 1$.

For each $\mathbf{u}, \mathbf{v}$ let $f(\mathbf{u}, \mathbf{v})$ be the vector

$$f(\mathbf{u}, \mathbf{v}) = \sum_{i=1}^{m} \alpha_i (\mathbf{u}^\top \mathbf{x}_i)(\mathbf{v}^\top \mathbf{x}_i) \mathbf{x}_i.$$

First, we claim that $f(\mathbf{u}^*, \mathbf{v}^*) \propto \mathbf{w}^*$ and that $F(\mathbf{w}^*, \mathbf{u}^*, \mathbf{v}^*) = \|f(\mathbf{u}^*, \mathbf{v}^*)\|$. Indeed for every $\|\mathbf{w}\| \leq 1$, by the Cauchy-Schwartz inequality:

$$F(\mathbf{w}, \mathbf{u}^*, \mathbf{v}^*) = f(\mathbf{u}^*, \mathbf{v}^*)^\top \mathbf{w} \leq \|f(\mathbf{u}^*, \mathbf{v}^*)\| \|\mathbf{w}\| \leq \|f(\mathbf{u}^*, \mathbf{v}^*)\|.$$

Again by Cauchy-Schwartz, equality is attained if and only if $\mathbf{w} \propto f(\mathbf{u}^*, \mathbf{v}^*)$.

Next, let us consider a single random variable $\hat{\mathbf{w}}$ such that $\mathbf{w} \sim N(0, Id)$ and $\hat{\mathbf{w}} = \frac{\mathbf{w}}{\|\mathbf{w}\|}$. Note that for any unit vector $\mathbf{u}_1$ we have $\mathbb{E}((\hat{\mathbf{w}}^\top \mathbf{u}_1)^2) = \frac{1}{d}$. Indeed, extend $\mathbf{u}_1$ to an orthonormal basis $\mathbf{u}_1, \ldots, \mathbf{u}_d$. we have that

$$1 = \mathbb{E}(\|\hat{\mathbf{w}}\|^2) = \mathbb{E}(\sum_{i=1}^{d}(\hat{\mathbf{w}}^\top \mathbf{u}_i)^2).$$

By symmetry we have that:

$$1 = \mathbb{E}(\sum_{i=1}^{d} (\hat{\mathbf{w}}^\top \mathbf{u}_i)^2) = \sum_{i=1}^{d} \mathbb{E}((\hat{\mathbf{w}}^\top \mathbf{u}_i)^2) = d\mathbb{E}((\hat{\mathbf{w}}^\top \mathbf{u}_1)^2).$$

In particular we have $\mathbb{E}((\hat{\mathbf{w}}^\top \mathbf{w}^*)^2) = \frac{1}{d}$. In conclusion $(\hat{\mathbf{w}}^\top \mathbf{w}^*)^2$ is a random variable that takes values in $[0, 1]$ and has expected value $\frac{1}{d}$. Applying the inverse Markov inequality (i.e. applying Markov to the random variable $1 - (\hat{\mathbf{w}}^\top \mathbf{w}^*)^2$), we have that

$$P((\hat{\mathbf{w}}^\top \mathbf{w}^*)^2 > \frac{1}{2d}) \geq \frac{\frac{1}{d} - \frac{1}{2d}}{1 - \frac{1}{2d}} = \frac{1}{2d - 1} \in O(\frac{1}{2d})$$

Letting $s \geq -\frac{\log \frac{1}{\delta}}{\log(1 - \frac{1}{2d})} \approx 2d \log \frac{1}{\delta}$ we have that with probability at least $(1 - \delta)$ for some $\mathbf{w}_i$ we have $|\mathbf{w}_i^\top \mathbf{w}^*| \geq \frac{1}{\sqrt{2d}}$, say $\mathbf{w}_1$.

Finally note that by definition of $\mathbf{u}_i, \mathbf{v}_i$:

$$\max_{i \leq s} F(\mathbf{w}_i, \mathbf{u}_i, \mathbf{v}_i) \geq F(\mathbf{w}_1, \mathbf{u}_1, \mathbf{v}_1) \geq (1 - \tau) \max_{\mathbf{u}, \mathbf{v}} F(\mathbf{w}_1, \mathbf{u}, \mathbf{v}) \geq (1 - \tau) F(\mathbf{w}_1, \mathbf{u}^*, \mathbf{v}^*) =$$

$$= (1 - \tau)\|f(\mathbf{u}^*, \mathbf{v}^*)\|\mathbf{w}^{*\top}\mathbf{w}_1 \geq \frac{1 - \tau}{\sqrt{2d}} f(\mathbf{u}^*, \mathbf{v}^*) = \frac{1 - \tau}{\sqrt{2d}} F(\mathbf{w}^*, \mathbf{u}^*, \mathbf{v}^*)$$

$\square$