[Reviews · NeurIPS 2014]

Submitted by Assigned_Reviewer_12

Summary: The paper provides an overview of what is known about the computational complexity of learning neural networks and contributes some news results. The authors suggest three considerations for making learning-theoretic progress on neural networks: over-specification, regularization, and using different activation functions. Using over-specification is mostly left for future work. Some negative results on using regularization and some positive results on using a squared activation function are presented.

Quality: Seems technically sound, though all the proofs were not checked in detail.

Clarity: Very clearly written and well-organized. Almost no typos.

Originality: A good number of new results, though some of these are straightforward applications of existing work. The paper also provides a nice summary of some of the existing positive and negative results for neural network learning.

Significance:

* Thm 2 is not particularly exciting –– the fact that the local minima are also global minima essentially means the network is uninteresting. The authors may believe that “the idea of over-specification is interesting and may become useful under some distributional assumptions,” but Thm 2 does nothing to further the goal of showing that over-specification is useful in any meaningful way.
* In §3, the paper considers the PAC learning model. The paper promises a “modern perspective” on NN learning, but the PAC framework is hardly “modern” –– it is a very strong notion of learnability, which leads to the many negative results given in the paper and others elsewhere. Considering, e.g., PAC-Bayesian approaches that might lead to positive learning results would be a much more productive research avenue.
* Cor 2 is not too surprising –– it is quite natural to need to regularize the l1 norm to be o(d). Nevertheless, this is a more interesting (and useful) result.
* §4 on polynomial networks (PNs) was the strongest portion of the paper.
* It is noteworthy that PNs are as powerful as threshold networks in the sense of expressing all TMs that run in time T using poly(T) nodes. However, this also means that most of the same hardness results will apply to PNs.
* Nevertheless, polynomial networks are quite conducive to theoretical analysis, as the authors demonstrate in Thm 4. Polynomial networks allow for the uniform approximation of other activation functions on a bounded interval using Chebyshev polynomials. This could be a promising approach to analysing neural nets under conditions of greater practical interest. I would encourage the authors to say more about the broader implications of PNs for future theory work.
* Regarding Thm 7, the paper should include a comment on the implications of choosing f \in P_{2,r} instead of \in P_{2,k} (how does this affect learning-theoretic guarantees, e.g.?).
* The paper only considers learning 2-layer PNs, not “deep” NNs (Although in the introduction the paper claims to learn depth-3 PNs, but only a subset, the class P_{3,k} of 3rd degree polynomials, is learned. This point should be corrected in the introduction).
* I encourage the authors to comment on the related pre-print Livni et al., “An Algorithm for Training Polynomial Networks” (arxiv.org/abs/1304.7045). How does the approach in this paper compare to the Livni et al approach?
* Can this paper’s approach to learning PNs be applied in depth (or degree) 4 or greater case? Please comment on whether approximate maximization of such higher order tensors might be possible.
* The experiments sections was very short and not especially insightful. Neural nets using sigmoid activation are never compared. Details of how data was collected is lacking. The authors claim that GECO is a good alternative to SGD, but the experiments are quite far from how deep learning is used in practice. I think the paper would be improved by simply removing §5 and including more exposition of the results and possible directions for future work.

Other comments:
line 149: “think on the network” => “think of the network”
lines 346-8: I couldn’t parse the sentence that begins “The vector w that...”

ADDENDUM:

Thank you for providing such a thorough response to the reviewer's questions and concerns. I think this is a solid paper that will make a nice addition to the literature on the computational and learning properties of NNs.
Summary: A solid theory paper that presents some tools and results that are likely to be useful in future neural net learning theory research.

Submitted by Assigned_Reviewer_15

The paper considers computational and theoretical questions around training multi-layer neural networks. In particular, the authors consider, in some restricted settings, when and how efficiently good solutions may be obtained. Both positive and negative results are presented. The identification of appropriate settings under which a useful analysis can be undertaken is itself a result.

Overall, the paper is clear and well-structured. Of the theory literature around neural networks, this effort is among the more grounded in reality, and more accessible. The paper includes several results which “sort of” flow together, giving the slightest hint of a manuscript that collected partially related pieces together into a whole. But, on balance the work is coherent enough. This reviewer did not check proofs in the supplementary material, however the general reasoning and flow of ideas presented in the paper itself appear to be sound.

The theoretical results do seem interesting on the whole, although Thm 2 and Cor. 1 may be subsumed in existing results, or may be incremental in nature. The authors spend a good deal of time studying networks induced by the quadratic activation function, which is non-standard, however they crucially try to understand how networks with this nonlinearity might be compared to networks defined from more widely adopted nonlinearities (such as the sigmoid).

Experimental validation is lacking: the authors ran out of space, and experiments are more of an afterthought. In a sub-field that is primarily driven by practical applications and experimental successes, experiments should probably feature prominently. One of the main contributions and selling-points of the paper is the “efficient algorithms” for training certain depth-2 and -3 networks, but it is hard to evaluate these methods. Limited experimental detail is provided, and only depth-2 networks are explored in a narrow context. The evidence given, however, appears positive. Weaknesses of the algorithms are not discussed in sufficient detail. More information about the experimental comparisons should be given, even if in the supplementary material, so that an interested reader can understand whether the baseline networks are realistic.

In the absence of more information and a sharpened impact message/result, the “practical” algorithms suggested by the theory may be of limited use. Inertia in this field seems strong, and many researchers might very well happily continue to use networks defined by sigmoid and rectification nonlinearities trained with vanilla SGD, at any depth they prefer. In this sense, the theory results may be viewed as the dominant contributions by which the paper may be evaluated.

Specific Comments:

- Section 2: What are your assumptions on the network, and the basic properties of the map f? The general mathematical setting lacks definition.
- The assumption that f(V) is continuous is also somewhat strong. Why do you need this?
- Theorem 2 says that the net has to have at least as many hidden units as there are training points, opening the door to obvious memorization of the data. What is the practical usefulness of this theorem? Combined with the (brief) experiments in Sec 5, is this saying that we’re making it easier to memorize the data?

- pg. 5, line 220: The classes of nnets defined by sigmoid and rectification nonlinearities are (proper) supersets of the \sigma_{0,1} class, so concluding from Corollary 1 that the former aren’t efficiently learnable doesn’t sound entirely helpful. Isn’t it precisely the networks that *aren’t* in \sigma_{0,1} which make the other two classes interesting? What can we say about, for instance, N_{\sigma_{sig}} \setdiff N_{\sigma_{0,1}} ?

- pg. 5, line 227: In what sense is N_{\sigma_{sig}, L} less “expressive” than N_{\sigma_{0,1}} ? In the sense of universal approximation networks, or the narrower sense where the former is supposed to include the latter as L\to\infty?

- Theorem 6: Does the result hold for any, unbounded n ? Don’t you need an asymptotic constraint on n dependent on d ?

- Sec 4.3 seems reminiscent of Oja’s algorithm (Oja flows). How can it be compared to that literature?

- More discussion about the computational complexity of the “modified” GECO algorithm is needed. In particular, the trace maximization required at each iteration, seems to potentially itself require many iterations (especially when there are many data points, or the problem lacks conditioning).

- Experiments Section: The second experiment (overspecification) is lacking. Does the plot show the training error (MSE)? If so, it doesn’t seem surprising that a network with more units is able to reduce the training error faster -- but perhaps this experiment is meant to confirm the expected? One could argue that it is easier for the network to memorize the data. More importantly, overspecification typically goes hand-in-hand with some form of regularization, as the goal is not to minimize train set error as quickly as possible, but to quickly reach a solution with good generalization performance. It would help if the experiment could address this key question as it’s central goal, with confirmation that the train error drops suitably fast as more of a corollary.
Summary: A collection of a few potentially interesting theory results around multi-layer (mainly 2,3) neural networks, presented in an accessible form. But possible questions about practical impact remain, and experiments are lacking.

Submitted by Assigned_Reviewer_47

Motivated by the proliferation of deep learning, this paper provides a collection of positive and negative results on the computational efficiency of training neural networks. The main results are:

(a) They show that any neural network that is significantly over-specified can be trained efficiently. This result is not terribly interesting, as it requires the number of neurons n to be greater than or equal to the number m of training examples.

(b) Extending results from Klivans and Sherstov, they show that the class of neural networks with t layers and n neurons is not efficiently learnable, even for super-constant n and t at least 2.

(c) They show that neural networks with bounded weights are as expressive as Turing Machines.

(d) Finally, they establish some hardness results, and introduce a new algorithm for learning 2-layer and 3-layer polynomial neural networks. The hardness results appear to be fairly easy extensions of existing methods, but the algorithmic results are quite interesting.

My opinion is that there are positives and negatives to this paper. On the positive side, the paper studies a very relevant and timely topic, and provides several computational complexity results which are not widely known. The main negative is that many of the results are small extensions of existing results in computational complexity. However, the contribution here is not so much the theoretical novelty, but more the exposition: the computational complexity results are re-interpreted for the NIPS community. As a result, I would recommend acceptance.
Summary: There are positives and negatives to this paper. The positive is that the paper studies a very timely topic, and provides interesting results that are not known to the NIPS community. The negative is that most of the results are fairly easy extensions of existing computational complexity result. Still, I recommend acceptance for its expository value.
Author Feedback
Author rebuttal: We would like to thank the reviewers for their comments and suggestions.
As the reviewers point out the paper revisit results on the computational complexity of deep learning as well as present some new non-trivial results. The results are not necessarily difficult but with current growing interest in a theory of deep learning we believe that this paper will be beneficial and helpful for researchers in the field.

As to some specific questions raised by the reviewers:

With respect to "modern perspective":

We put less emphasis on sample complexity analysis and more emphasis on computational issues. By modern perspective we were referring to things like over-specification and ReLU's, which are not necessarily covered by classical NN results.

The paper only considers learning 2-layer PNs, not “deep” NNs:

We will clarify the point in the introduction to avoid misunderstanding. That being said, training 2-layer PNs poses a non-convex problem that cannot be addressed by "flattening" or "linearization" (without a significant payoff in sample complexity). In that sense our algorithm suggests a non-trivial solution to a problem posed by a deep learning architecture.

Can this paper’s approach to learning PNs be applied in depth (or degree) 4 or greater case?

An extension to higher degrees is possible, though currently it produces r of order O(d^(m-2)) where m is the degree. This is not an impressive improvement for m>4 and we've decided to neglect this direction in current work.

Neural nets using sigmoid activation are never compared:

We've compared sigmoid activation function and found the ReLu activation function to be superior in our setting: therefor we presented only the ReLu results. Further, ReLU is the more standard practice nowadays, rather than sigmoids (E.g. Krizhevsky et al.'s imagenet paper)

The assumption that f(V) is continuous is also somewhat strong:

As we point out the assumption holds whenever the activation function is continuous. Currently the assumption is crucial for our proof (see line 492 in appendix).

pg. 5, line 220:

The reviewer is correct in his remark. We address the point raised by the reviewer in the next paragraph (line 224). We discuss regularized nnets as they exclude nnets that are in \sigma_{0,1} (specifically, they exclude intersection of half spaces). As we show in corollary 2, these are still hard to learn.

In what sense is N_{\sigma_{sig}, L} less “expressive” than N_{\sigma_{0,1}} ?

In the latter, restricted, sense. That for fixed L we can only approximate \sigma_{0,1}. We will rephrase the statement here and make it more accurate.

Thm 6:

No need to bound n. Our proof relies on an approximation of sigmoidal via a polynomial- very roughly it leads to an embedding of the neurons in a finite dimensional space which leads to bounds that do not depend on number of neurons (and depends on the dimension).

Sec 4.3 seems reminiscent to oja's flow:

Though the reviewer's remark seems to be interesting, we have to admit that the connection escapes us. We will be happy to hear more of it.

We will also clarify the statements in thm 7 and improve section 2 as suggested by the reviewers, as well as address other improvements suggested by the reviewers.